# Passive Backscatter Communication Scheme for OFDM-IM with Dynamic Carrier Activation

**DOI:** 10.3390/s23083841

**Published:** 2023-04-09

**Authors:** Shibao Li, Rui Lu

**Affiliations:** College of Oceanography and Space Informatics, China University of Petroleum (East China), Qingdao 266580, China

**Keywords:** backscatter communication, OFDM, index modulation, bit error rate

## Abstract

Multicarrier backscattering has been proposed to improve the communication rate, but the complex circuit structure of multicarrier backscattering devices requires more power consumption, resulting in devices far away from the radio frequency (RF) source without enough power to maintain communication, which greatly reduces the limited communication range in backscattering. To solve this problem, this paper introduces carrier index modulation (IM) into orthogonal frequency division multiplexing (OFDM) backscattering and proposes a dynamic subcarrier activated OFDM-IM uplink communication scheme suitable for passive backscattering devices. When the existing power collection level of the backscatter device is detected, only a subset of carrier modulation is activated using part of the circuit modules to reduce the power threshold required for device activation. The activated subcarriers are mapped by a block-wise combined index using the look-up table method, which can not only transmit information using traditional constellation modulation but also carry additional information through the frequency domain carrier index. Monte Carlo experiments show that this scheme can effectively increase the communication distance and improve the spectral efficiency of low-order modulation backscattering when the power of the transmitting source is limited.

## 1. Introduction

The next-generation wireless network needs to support various Internet of Things services, and some scenarios have the characteristics of low power consumption, delay tolerance, and large-scale deployment [1]. Backscatter communication uses passive backscatter devices capable of modulating their messages via incident sinusoidal carriers or ambient RF carriers without the use of power-hungry and expensive RF transmitters. It is an energy-saving and low-cost communication technology used for the Internet of Things [2].

The reason why the backscatter design maintains a low communication capacity is that the spectral efficiency is low under the single carrier modulation scheme [3], so multicarrier technology is applied to the backscatter to improve the network capacity. In reference [4], the excitation signal was reflected, modulated, and shifted by the tag to be located in the frequency band of the OFDM subcarrier, and the overall transmit power consumption of the implementation scheme was conservatively estimated to be 116.58 µW. In [5], several low-power design schemes were proposed for OFDM backscatter hardware design, and the backscatter modulator consumed 600 µW. Compared to single-carrier modulation backscatter technology, multicarrier backscatter devices have higher power requirements.

However, backscatter systems suffer from the double near–far effect caused by the large-scale fading of wireless communication channels [6], which leads to the inability of devices far away from the RF source to harvest enough energy to maintain the communication circuit. A tunnel diode-mounted FM band backscatter tag amplifier was designed and implemented in [7], which reduced the power threshold for tag activation and increased the transmission distance. In [8], an IRS-assisted relay transmission scheme was proposed to enable more energy to be transmitted to the remote end. In [9], by dynamically adjusting the transmit power of the base station and traversing the users one by one, the RF source power increased with an increasing distance between the device and the source node. With a completely asymmetric transmission protocol in [10], edge users used more energy-efficient modulation, which enabled them to support the power consumption of several microwatts to tens of microwatts. Based on the mathematical model of the distance between the transceiver and the backscatter node, beamforming was transmitted from the RF source in [11] to improve the effect of energy harvesting for long-distance users. In order to provide device circuit consumption, backscatter based on energy harvesting has been proposed. The device needs to collect energy for a long time to achieve the activation condition, which cannot complete real-time communication [12,13,14]. Due to the energy efficiency advantage of index modulation, the BackCom system based on energy balance has been proposed [15]. The existing IM schemes that support BackCom [16,17] proceed mainly with the help of antenna indexing in space, carrying index information by activating different antennas.

The above literature has its own advantages, but the OFDM modulation structure of multicarrier equipment was not considered, so it is not applicable to multicarrier backscattering equipment. For this purpose, the OFDM technique with index modulation is combined with backscattering to balance the power consumption of the device. By analyzing the power collection and energy consumption of the equipment, an OFDM-IM model based on the number of distance-adaptive activated subcarriers is proposed, and the strategy of maximizing the spectral efficiency activated subcarriers is adopted to further improve the communication quality of remote users.

## 2. Model of OFDM Backscatter System

The design of the backscatter device that can perform OFDM modulation is shown in Figure 1. The device consists of an antenna, N single-sideband modulation (SSB) modules, and the corresponding impedance circuit. Each single sideband modulation module can simulate and generate an OFDM subcarrier. The continuous sinusoidal wave sent by the RF source is divided into 2N channels in the backscatter device, and each two channels are modulated as a group for SSB. Multicarrier backscatter devices can be assigned N subcarriers, so that, theoretically, the data rate is N times higher relative to single-carrier modulation and, in practice, can be applied to devices with higher bandwidth requirements.

For a single subcarrier modulation module, when it controls the backscattered carrier signal with a frequency Δf switch, it essentially uses a square wave to modulate the phase of the sample. Using $\theta_{n}$ to generate the phase information, the square wave $W_{n}$ can be expressed using a Fourier series as follows:(1)$$W_{n}\left(\Delta f,\theta_{n}\right)=0.5+\frac{2}{\pi }\overset{\infty }{\underset{m=1,3,5,\ldots }{\sum }}\frac{1}{m}\cos\left(2\pi m\Delta ft+\theta_{n}\right).$$

The modulated square wave signal is multiplied by a continuous sine wave to produce a backscattered symbol. For the *n*th subcarrier, the backscattered sample can be calculated by multiplying the ambient carrier $S_{cw}$ with the first harmonic of the square wave as follows:(2)$$\begin{array}{ll} x_{n} & =\frac{2}{\pi }S_{cw}\cos\left(2\pi n\Delta ft+\theta_{n}\right)\\ & =\frac{1}{\pi }\left\{S_{cw}e^{j\theta_{n}}e^{j2\pi n\Delta ft}+S_{cw}e^{j\theta_{n}}e^{j2\pi n\Delta ft}\right\}. \end{array}$$

It is worth noting that the excited subcarrier generates two side bands of symmetry through the modulation and frequency shift of the square wave. As described by HitchHike [18], the useless sideband can be easily eliminated by making a negative copy of the signal on the useless sideband. For a device equipped with multiple SSB modules, the reflected signal is the sum of the signals of multiple submodules, which can be expressed as:(3)$$x=\frac{1}{\pi }S_{cw}\overset{N}{\underset{n=1}{\sum }}e^{-j\theta_{n}}e^{-j2\pi n\Delta ft}.$$

The backscatter device is internally composed of N energy-consuming single sideband modules and corresponding clock and processing cores, so the total power consumption of the transmitter circuit is the sum of the following:(4)$$P_{BD}=N\times P_{c}+N\times P_{SSB}+N\times P_{D},$$
where $P_{c}$, $P_{SSB}$and $P_{D}$ represent the clock module, single sideband modulation module, and digital switching power consumption, respectively. Let $P_{sc}=P_{c}+P_{SSB}+P_{D}$ denote the power consumption of a single subcarrier and assume that all backscatter emitters have the same power consumption.

The system we consider, as shown in Figure 2, consists of one carrier transmitter (CT) and U backscatter devices. The CT sends a continuous sine wave to provide the carrier and communication energy for the backscatter device and collects the backscatter information of the device. The nodes are uniformly distributed in a ring coverage area specified by the inner radius $R_{1}$ and outer radius $R_{2}$, where CT is located at the origin.

Taking into account the wireless fading channel model—that is, when the transmit power at the transmitter is $P_{t}$—the received power at the receiver is $P_{r}=P_{t}d^{-\alpha }$, where d is the distance between the transmitter and the receiver and α is the path loss exponent. This is a reasonable assumption for BackCom systems with strong line-of-sight (LOS) links. This scheme is still applicable to the conventional channel model, and $P_{r}=P_{t}{\left|h\right|}^{2}$ at the receiver for the system with a channel gain *h*. We denote the size of the reflection coefficient of device u as $\varepsilon_{u}$, where $0\leq \varepsilon_{u}\leq 1$. Part of $P_{r}$ will be harvested by the energy harvester and provide the circuit work of device u. This part of the power is given by [19]:(5)$$\begin{array}{ll} P_{H,k} & =\rho \left(1-\varepsilon_{u}\right)P_{r,u}\\ & =\rho \left(1-\varepsilon_{u}\right)P_{t}d_{u}{}^{-\alpha },\forall u\in U, \end{array}$$
where ρ represents the energy collector efficiency, which is between 0 and 1. This part of the power is utilized by the backscatter device to maintain the backscatter circuit operation to modulate and reflect the incident carrier signal. The remaining part of the incident signal power $P_{xu}=\varepsilon_{u}P_{ru}$ is used for the signal transmission of device *u*. The backscatter emitter does not require a battery and cannot store the harvested energy. We assume that the energy harvested by the transmitter is only used to support the circuit operation of the current symbol [20,21]—that is, $P_{Hu}\geq P_{BD}$, where $P_{BD}=\overset{N}{\underset{n=1}{\sum }}P_{sc}$ is the device activation power threshold.

## 3. Backscatter OFDM-IM Scheme

### 3.1. Backscatter Modulation

For a device with a fixed reflection coefficient, whether it can activate communication is only related to its relative position in terms of the RF source and its own power threshold; therefore, part of the circuit can be activated according to different power-harvesting levels. As a result of the partial activation of the subcarriers, it is an effective practice to add OFDM-IM to improve its spectral efficiency.

In the OFDM-IM system, the activated subcarrier index and the traditional constellation symbol are used to transmit information. As shown in Figure 3, for an OFDM-IM block with allocated $N_{u}$ subcarriers and *k_u_* active subcarriers, the transmitted $b_{u}$ bits are divided into two parts.

First, the index bits $b_{u,idx}=\lfloor \log_{2}(C(N_{u},k_{u}))\rfloor$ look-up tables are used to determine the subcarrier $k_{u}$ subblock of activity, and the activity subcarrier index set is set to $I_{u}(\xi )=\{i_{u,1}(\xi ),...i_{u,ku}(\xi )\}$, where ⌊∙⌋ denotes the rounded-down function. The $b_{u,sym}=k_{u}\log_{2}M_{u}$ bit by $q_{u}(\xi )=[q_{u,1}(\xi ),...q_{u,ku}(\xi )]$ M-ary modulates information through the activity of subcarrier transfer, where $q_{u,k}\left(\xi \right)\in Q_{u}$, and $Q_{u}$ is the complex signal constellation diagram of size $M_{u}$ for device u and normalized to the unit average power. Therefore, in total:(6)$$\begin{array}{ll} b_{u} & =b_{u,idx}+b_{u,sym}\\ & =\lfloor \log_{2}\left(C\left(N_{u},k_{u}\right))\right\rfloor +k_{u}\log_{2}M_{u}, \end{array}$$
bits are sent by the index block. When $k_{u}=N_{u}$, the system backs off from OFDM-IM to conventional OFDM modulation. For each OFDM-IM signal, ***X*** is jointly created by $I_{u}\left(\xi \right)$ and $q_{u}\left(\xi \right)$.

### 3.2. Subcarrier Activation Scheme

In this case, we use the variable r to denote the distance from the CT to the backscatter device; then, its distribution can be expressed as $f\left(r\right)=2r/\left(R_{2}^{2}-R_{1}^{2}\right)$. As shown in Equation (4), the backscatter activation power has a linear relationship with the multicarrier capability. To improve the coverage of CT, an OFDM-IM scheme with dynamic subcarrier activation is proposed. Specifically, the backscatter device considers each subcarrier modulation module individually and, then, depending on the strength of its energy harvesting, dynamically activates part of the circuit to reduce the backscatter circuit power threshold. Due to its low-power property, each backscatter device is equipped with only a finite number of subcarriers, and we consider all carriers of a device as an index symbol block. Thus, the number of reflected subcarrier activations n can be determined as follows:(7)$$n\leq \frac{\rho \left(1-\varepsilon_{u}\right)P_{r,u}}{P_{sc}},n\in \left[1,2,\ldots ,N_{u}\right].$$

Therefore, the coverage area is divided into $N_{u}$ concentric rings, and the maximum number of active subcarriers in each ring is n. The probability of the number of subcarriers n activated by the device within the coverage range is given:(8)$$\begin{array}{ll} P\left(n\right)\& & =\int_{r_{n+1}}^{r_{n}}f\left(r\right)dr=\int_{r_{n+1}}^{r_{n}}\frac{2r}{R_{2}^{2}-R_{1}^{2}}dr\\ & =\frac{r_{n}{}^{2}-r_{n+1}^{2}}{R_{2}^{2}-R_{1}^{2}},n\in \left[1,2,\ldots ,N_{u}\right], \end{array}$$
where $r_{n}$ represents the maximum radius with n number of activated subcarriers, the special $r_{N+1}=R_{1}$ is the inner radius, and $r_{n}$ can be determined by the following formula:(9)$$r_{n}=\sqrt[{\alpha }]{\frac{\rho \left(1-\varepsilon \right)P_{t}}{nP_{sc}}},n\in \left[1,2,\ldots ,N_{u}\right].$$

The range gain of the proposed scheme is $\Delta r=r_{1}-r_{N}$ with respect to the conventional OFDM modulation.

The information sent by an OFDM-IM symbol is composed of two parts: one is the index bit and the other is the symbol bit. For device u with subcarrier block $N_{u}$ and active subcarriers $k_{u}$, the spectral efficiency can be calculated as follows:(10)$$\eta_{SE,k_{u}}=\frac{\lfloor \log_{2}\left(C\left(N_{u},k_{u}\right))\right\rfloor +k_{u}\log_{2}\left(M_{u}\right)}{N_{u}}\left[\mathrm{bps}/\mathrm{Hz}\right]$$

It can be seen from Equation (10) that, for the backscattering device u, the constellation modulation order $M_{u}$, the number of carriers in a single subblock $N_{u}$ and the number of active carriers $k_{u}$ in a subblock will directly affect the spectral efficiency of the system without considering the influence of the length of the cyclic prefix.

In backscatter, due to hardware limitations, low-order modulation and a limited number of subcarriers are usually used, which makes the spectral efficiency of index modulation not lower than that of full carrier activation in conventional cases. For $k_{u}$ = 1, 2, and 3, in the BPSK system used, the spectral efficiency is 3/4, 1, and 5/4, respectively, which is 1 compared to the ${\bar{\eta }}_{SE}$ of traditional OFDM. It can be seen that, under low-order modulation, partial carrier activation can bring higher spectral efficiency. Therefore, the device can choose the optimal carrier activation scheme according to the modulation modes it supports, where the number of activated carriers is $k_{u}\leq n$.

After determining the number of carrier activations, the carrier activation mapping is completed by looking up the mapping table. Table 1 shows a carrier mapping mode with $N_{u}=4$ and $k_{u}=2$.

If the data stream $b_{idx}=\left[0,\;1\right]$, the device activates the subcarriers with index numbers 2 and 3 by querying the mapping table, and the signal subblock after mapping is ${\left[0,\;q_{1},\;q_{2},\;0\right]}^{T}$. This mapping table is not fixed, and some combinations are discarded, since the number of combinations does not satisfy an integer power of two. To further reduce the system error, the index map with the minimum Euclidean distance is discarded. The Euclidean distance is defined as follows:(11)$$d_{min}\left(H\right)=\underset{\begin{array}{c} X_{i},X_{j}\in X\\ X_{i}\neq X_{j} \end{array}}{\min}||H\left(X_{i}-X_{j}\right)||,$$
where $X_{i}$ and $X_{j}$ denote the symbols in the M-PSK modulation constellation, and H is the channel fading matrix. For simplicity, we can complete the index mapping by reducing the use of subcarrier combinations with smaller channel gains. For example, for subcarrier channel H:(12)$$H=\left[\begin{array}{cc} \begin{array}{cc} 1.576+0.238i & 0.707-1.176i\\ 0.337-0.857i & 0.046+0.461i \end{array} & \begin{array}{cc} -0.417-0.196i & 0.298-1.181i\\ 0.231+0.765i & 0.712-0.462i \end{array}\\ \begin{array}{cc} 0.186-0.667i & -0.934+0.653i\\ 0.247+0.884i & 0.657+0.1695i \end{array} & \begin{array}{cc} 0.352-0.038i & 0.644+0.421i\\ -0.488-0.464i & 0.842-1.139i \end{array} \end{array}\right]$$
we should abandon the combination of {1, 3} and {3, 4}.

### 3.3. Signal Detection

At the backscatter end, the information bits are divided into index bits and modulation bits, which are integrated into OFDM-IM subblocks and received by the detector after passing through the reflection channel. The received signals are as follows:(13)Y=εHX+n
where *n* is Gaussian additive noise subject to a zero mean and variance $\sigma^{2}$.

The maximum likelihood (ML) detection algorithm is the most commonly used method for the receiver to detect the correct sending signal from the output signal. It mainly searches all possible signals through traversal; that is, it uses this exhaustive search to detect index combinations and modulation symbols and seeks to minimize the Euclidean distance between index subcarriers and modulation symbols, which are the most likely index subcarriers and constellation symbols of the system at this moment.
(14)$$\hat{x}=\underset{x\in \chi }{\arg\min}||Y-\varepsilon \hat{H}X{||}^{2}$$
where $\hat{H}$ represents the estimated channel vector and χ represents all possibilities of transmitting vector x, including the activation subcarrier index value and constellation symbol. To confirm the mapping table, the carrier activation information should be exchanged before demodulation, and the overall complexity is $O\left(2^{{\lfloor \log}_{2}\left(C\left(N,k\right))\right\rfloor }M^{k}\right)$.

## 4. BER Analysis

In the process of information transmission, the Euclidean distance of constellation points determines the system error performance. Due to the sparsity of carriers in the OFDM-IM system, the Euclidean distance of the edge signal can be increased, because the fewer activated subcarriers, the lower the number of mapping$\;2^{b_{u}}$ for index demodulation. At the same time, as shown in Figure 4, the fewer subcarriers are activated, the power of the inactive subcarriers spreads over the active subcarriers, and the energy $E_{s}$ for the M-ary symbol increases by a factor of $N_{u}/k_{u}$ times.

The upper bound of the BER for OFDM-IM is obtained from the joint-bind technique. The union bound means that, for any finite or countable set of events, the probability that at least one of the events occurs is not greater than the sum of the probabilities of the individual events, as shown in the following equation:(15)$$P_{e,u}\leq \frac{1}{b_{u}2^{b_{u}}}\underset{\begin{array}{c} X,\hat{X}\\ ℐ\neq \hat{ℐ} \end{array}}{\sum }\mathrm{Pr}\left(X\to \hat{X}\right)G\left(X,\hat{X}\right),$$
where $G\left(X,\hat{X}\right)$ is the number of error bits when X is detected as $\hat{X}$. For unconditional pairwise error probability $\mathrm{Pr}\left(X\to \hat{X}\right)$, said transmission symbol X is the probability of detection for $\hat{X}$ wrongly:(16)$$\mathrm{Pr}\left(X\to \hat{X}|H\right)=Q\left(\sqrt{\frac{\vartheta }{2}||(X-\hat{X})H{||}^{2}}\right),$$
where $\vartheta =1⁄\sigma^{2}$ is the average signal-to-noise ratio (SNR) of each subcarrier. By the Q function approximation of $Q\left(x\right)\approx 1/12\cdot e^{-x^{2}/2}+1/4\cdot e^{-2x^{2}/3}$, unconditional pair-wise error probability [22] can easily be, from Equation (16), deduced as:(17)$$\begin{array}{c} \mathrm{Pr}\left(X\to \hat{X}\right)≅E_{H}\left\{\frac{e^{-\frac{E_{s}}{4\sigma_{o}^{2}}}}{12}+\frac{e^{-\frac{E_{s}}{3\sigma_{o}^{2}}}}{4}\right\}\\ ≅\frac{1/12}{\det\left(I_{N}+\left(\rho /2\right)KR\right)}+\frac{1/4}{\det\left(I_{N}+\left(2\rho /3\right)KR\right)}, \end{array}$$
where $K=E_{H}\left\{H^{H}H\right\}$ is the covariance matrix of H and $R={\left(X-\hat{X}\right)}^{H}\left(X-\hat{X}\right)$. It can be seen that the system BER is related to the Euclidean distance between the received symbols, and fewer carrier activations make the symbol energy increase, which improves the BER performance of the system.

## 5. Simulation Results

In this section, the Monte Carlo method is used to verify the performance of the proposed system and compare it with the conventional OFDM system. As shown in Figure 2, the connectivity performance is reflected by the number of decoded backscatter nodes that are uniformly distributed, assuming that the readers are uniformly distributed over a large enough range. Part of the experimental data are given in Table 2.

Due to the use of a large number of digital logic cells, the traditional multicarrier backscatter scheme raises the backscatter device activation power threshold, which limits the deployment of backscatter devices. However, our proposed OFDM-IM scheme with dynamic subcarrier activation can flexibly enable part of the circuit to achieve low power consumption. The device is assumed to use as many activated carriers as possible for communication according to its energy harvesting. In Figure 5a, we compare the distribution of the number of devices in the system with different subcarrier activation levels. Compared with the traditional OFDM backscatter, the dynamic carrier activation method reduces the activation power threshold by activating fewer carriers, and the gain of the number of device activations is $\overset{N-1}{\underset{k}{\sum }}u_{k}$, where $u_{k}$ represents the number of devices using the most k subcarriers. In this scheme, the equipment that meets the traditional OFDM activation power can still use the traditional method to transmit. As can be seen in Figure 5b, with the increase of the transmit power of the excitation source, the communication distance of the system shows an exponential growth. When the transmit power of the excitation source is 30 dBm, the traditional OFDM backscatter device can complete the activation at 16 m at the farthest, while the OFDM-IM with a single subcarrier can complete the activation at 27 m at the farthest. We use the FPGA model of Xilinx ZYQN-7000 for the logic simulation. Without using memory, the average power consumption of OFDM-IM with single-carrier activation is 36.18 μW, compared with 118.42 μW of OFDM modulation with four-carrier activation. This benefits from the fact that partial carrier activation uses fewer logic cells and clock modules.

In backscatter communication, not only should the device activation power be considered but, also, the minimum SNR constraint for decoding at the receiver should be taken into account. The receiver decodes the signal SNR threshold set to *γ*. Only when the reflected signal SNR compared to *γ* signals can be decoded successfully. We define the SNR constraint as follows:(18)$$\frac{\varepsilon_{k}P_{t}d^{-2\alpha }}{P_{n}}\geq \gamma \;,$$
where $P_{n}$ is the ambient noise power. In the training phase, for devices in which SNR does not reach γ, the CT controls them to remain off, which helps to reduce inter-user interference and device energy consumption. It can also be seen in Figure 6 that, when the power at the transmitter is 35 dBm, the number of decoders gradually decreases with the increasing SNR threshold required for demodulation. The reason is that the reflected signal does not meet the threshold requirement and the reader cannot decode it. However, when the threshold value increases, the weak signal is no longer decoded. Then, the average number of decoders will be reduced. Once the threshold is extremely large, the average number of decoders and throughput will become zero; that is, the communication is interrupted. Devices using OFDM-IM perform well at low SNR demands, but the edge user decoding performance degrades rapidly as the SNR rises. This is because devices with a small number of activated carriers within uniformly distributed circular coverage are located at the edge of the RF source and experience double large-scale fading, resulting in severe attenuation of their reflected signal power.

In Figure 7, we explore the influence of the reflection coefficient on the number of system decodes under the joint constraints of the SNR and device activation power. The reflection coefficient divides the acquired energy into two parts: one for the label circuit energy supply and the rest for the backscattered signal transmission. As the reflection coefficient increases, the number of decodes gradually increases. At this stage, the increase in the reflection coefficient will reflect more power to achieve the desired γ of demodulation. After reaching the high point, the number of decodes dropped rapidly, because the high reflection coefficient prevented the device from collecting enough energy. At the same time, it can be seen that, with the increase of γ, the transformation of the optimal reflection coefficient is inversely proportional to γ. The reflection coefficient is concentrated between [0.8] and [0.9], which is related to the power consumption of the device.

In Figure 8, the BER performance of BPSK modulation with $k_{u}=1,\;2,\;\mathrm{and}\;3$ and conventional OFDM signals in the AWGN channel are considered, and the perfect channel state estimation is assumed in the simulations. As the SNR increases, the BER performance of the system gradually improves. It can be seen that, at the same SNR, OFDM-IM exhibits a better BER performance because of the lower-order modulation and the higher power allocated to a single modulation symbol. The use of lower order modulation and better error performance make OFDM-IM more suitable for backscatter device communication, and it can also achieve a lower complexity detector architecture and be more energy-efficient. For the case of N = 4, the SNR at 10^−4^ BER is about 26 dB, 29 dB, and 32 dB when the number of activated subcarriers k = 1, 2, and 3, respectively. According to Equation (10), the corresponding spectral efficiencies of the device are 3/4, 1, and 5/4, respectively, and it can be seen that the BER performance decreases by nearly 3 dB for every quarter of the spectral efficiency improvement. In other words, the system can activate fewer subcarriers and sacrifice part of the spectral efficiency to achieve reliable communication with low BER when the transmission power of the excitation source is fixed.

## 6. Conclusions

According to the structure of a multicarrier backscatter device, orthogonal frequency division multiplexing with index modulation is introduced into the backscatter for the first time in this paper, and the power consumption is reduced by using a partial circuit activation of index modulation. Compared with the traditional backscatter OFDM system, the performance of the proposed system is superior in terms of the bit error rate and transmission distance. In the case that each backscatter device contains the number of carriers N=4, the farthest activation distance can be improved about twice. In terms of BER, it is verified that, at low modulation orders, the OFDM-IM system with sparse carrier activation can bring lower BER compared with the traditional OFDM. In practice, the carrier can be flexibly activated to reduce the communication outage probability according to the device deployment distance and data reliability requirements. Compared with the traditional OFDM backscatter system, the device based on OFDM-IM can flexibly use the collected power to meet the purpose of long-distance transmission.

## Figures and Tables

**Figure 1 sensors-23-03841-f001:**
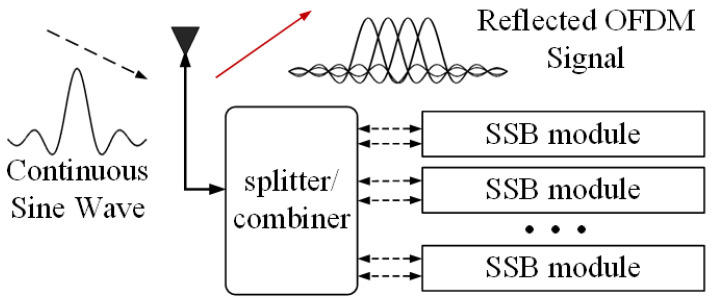
Structure of a multicarrier backscatter device.

**Figure 2 sensors-23-03841-f002:**
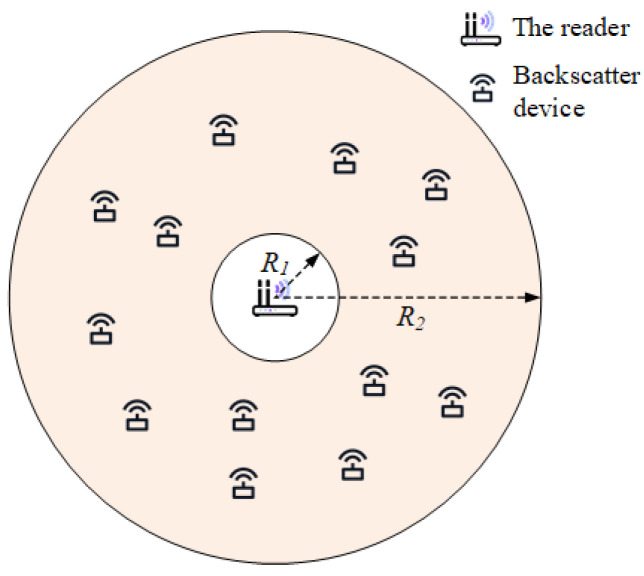
Spatial model in an annulus coverage of backscatter devices.

**Figure 3 sensors-23-03841-f003:**
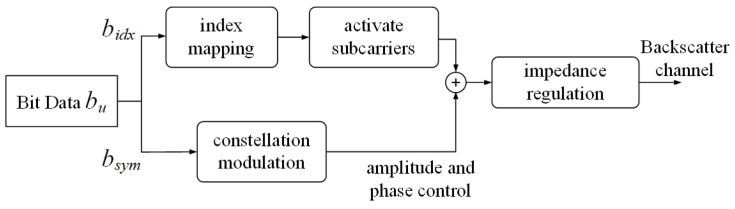
OFDM backscatter flow with index modulation.

**Figure 4 sensors-23-03841-f004:**
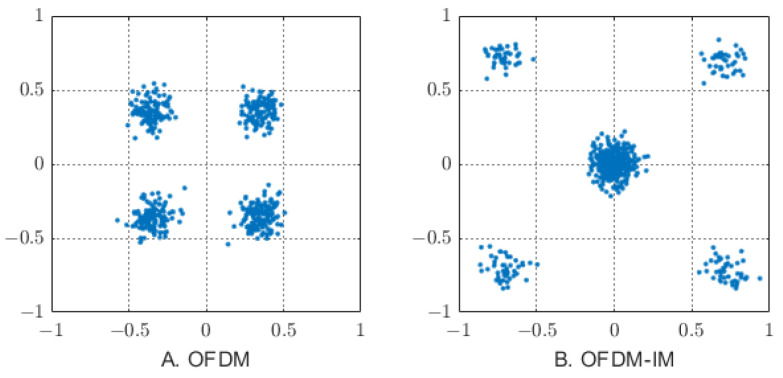
Comparison of the constellation diagrams between conventional (**A**) OFDM and (**B**) OFDM-IM.

**Figure 5 sensors-23-03841-f005:**
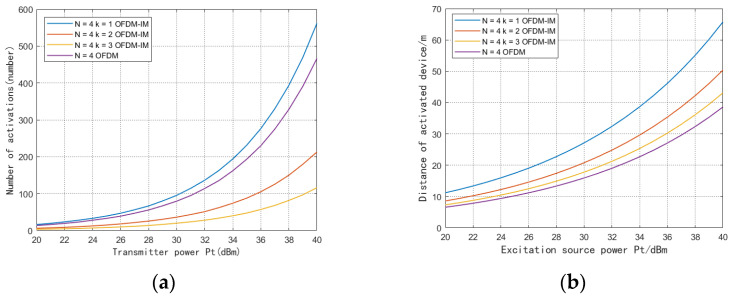
(**a**) Number of activations of different power devices at the transmitter. (**b**) Activation distance of different power devices at the transmitter.

**Figure 6 sensors-23-03841-f006:**
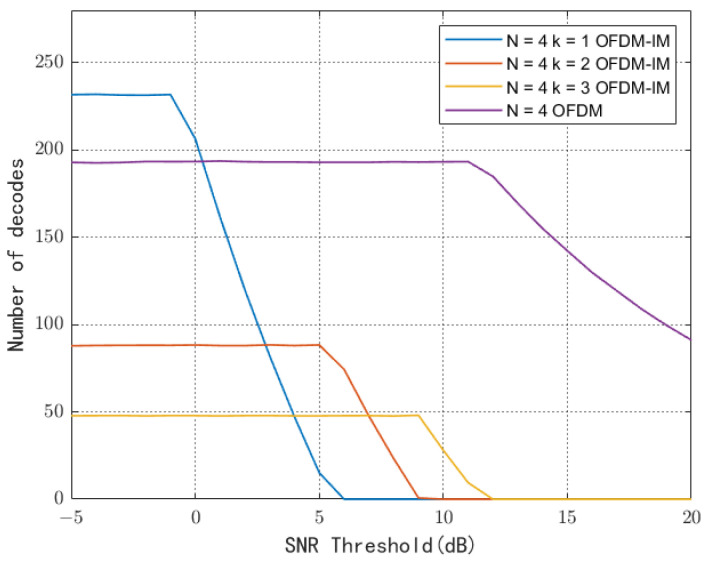
Decoding number comparison under different SNR.

**Figure 7 sensors-23-03841-f007:**
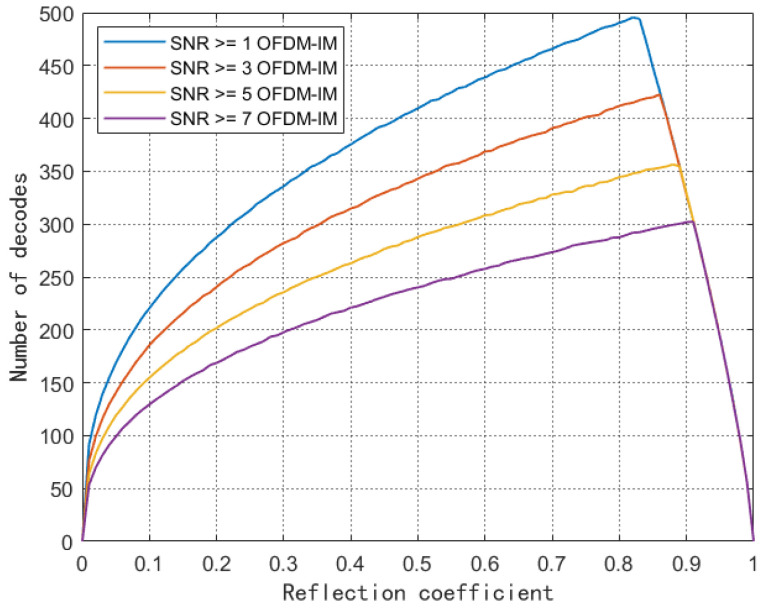
Reflection coefficient and number of device decodes.

**Figure 8 sensors-23-03841-f008:**
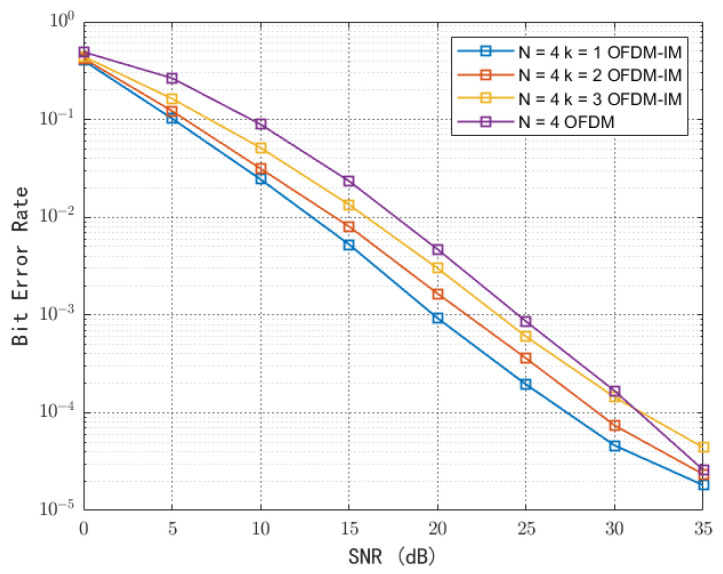
Bit rate of OFDM and OFDM-IM under BPSK modulation.

**Table 1 sensors-23-03841-t001:** The mapping table of the OFDM-IM scheme with $N_{u}=4$ and $k_{u}=2$.

Data Bit	Subcarrier Combination	Subcarrier Block
[0, 0]	{1, 2}	${\left[q_{1},\;q_{2},\;0,\;0\right]}^{T}$
[0, 1]	{2, 3}	${\left[0,\;q_{1},\;q_{2},\;0\right]}^{T}$
[1, 0]	{2, 4}	${\left[0,\;q_{1},\;0,q_{2}\right]}^{T}$
[1, 1]	{1, 4}	${\left[q_{1},\;0,\;0,\;q_{2}\right]}^{T}$

**Table 2 sensors-23-03841-t002:** Simulation experiment parameters.

Parameter	Value
density of device	$0.1\;\mathrm{num}/m^{2}$
number of subcarriers	4
path-loss-exponent	2.6
Energy conversion efficiency	0.8
Transmit power	20–40 dBm
noise power	−90 dBm

## Data Availability

The data presented in this study are available on request from the corresponding author.

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
