# Peer review of "Passive Backscatter Communication Scheme for OFDM-IM with Dynamic Carrier Activation"

_sensors, 2023, doi:10.3390/s23083841_

Round 1

Reviewer 1 Report

This paper proposes a solution to the limited communication range in backscattering by introducing carrier index modulation into OFDM backscattering. This results in a dynamic subcarrier activated OFDM-IM uplink communication scheme suitable for passive backscattering equipment. I think a general methodology is appropriate and makes sense in terms of backscattering communications.

What is trivial but important thing is that abbreviation should be defined at the beginning, e.g., OFDM-IM, though readers are likely to realize it.

The purpose of the paper seems to have a contribution of reduction in power consumption, however, the results of the simulation do not explain directly, but only indirect results. The paper needs to show a decrease in power consumption compared to the same environment for the same environment.

Therefore, I think minor revision is needed for the manuscript to be published.

Reviewer 2 Report

In this paper, the author introduces carrier index modulation into OFDM backscattering, and proposes a dynamic subcarrier activated OFDM-IM uplink communication scheme suitable for passive backscattering equipment. The Monte Carlo simulation results show that the scheme can effectively increase the communication distance and improve the spectral efficiency of low-order modulated backscatter when the power of the transmitting source is limited. The following questions need be addressed, and the format and grammar should be checked carefully .

1. In Section 3.2, the mapping relationship listed in Table 1 of the mapping relationship of the example data flow [0, 1] does not correspond, and should be {2,3} and [0,q1,q2,0]T.

2. Before analyzing the simulation results, the author should add the whole simulation system block diagram.

3. As far as I know, the traditional ML algorithm will bring high computational complexity by traversal search and screening possible signals. In section 3.3, the author also uses the ML algorithm and calculates the overall complexity. I suggest the author compare the computational complexity of different signal detection algorithms to prove the superiority of the proposed scheme.

4. In section 5, the author proves that the transmission distance of the proposed scheme is longer than that of the traditional OFDM system through Figure 4. Figure 4 cannot directly show the author's conclusion. It is suggested that Figure 4 more clearly compares the transmission distance between the proposed scheme and the traditional OFDM scheme.

5. In section 5, the author said that the proposed scheme can sacrifice part of the spectrum efficiency to improve the error rate performance of the system. Can the author quantify the loss of spectrum efficiency.
